

# Application of interval type-2 fuzzy logic and type-1 fuzzy logic-based approaches to social networks for spam detection with combined feature capabilities

İsmail Atacak[1], Oğuzhan Çıtlak[2] and İbrahim Alper Doğru[1]

[1] IoTLab, Department of Computer Engineering, Faculty of Technology, Gazi University, Ankara, Turkey
[2] Graduate School of Natural and Applied Sciences, Gazi University Central Campus, Faculty of Technology, Teknikokullar, Gazi University, Ankara, Turkey

## ABSTRACT

**Background**. Social networks are large platforms that allow their users to interact with each other on the Internet. Today, the widespread use of social networks has made them vulnerable to malicious use through different methods such as fake accounts and spam. As a result, many social network users are exposed to the harmful effects of spam accounts created by malicious people. Although Twitter, one of the most popular social networking platforms, uses spam filters to protect its users from the harmful effects of spam, these filters are insufficient to detect spam accounts that exhibit new methods and behaviours. That's why on social networking platforms like Twitter, it has become a necessity to use robust and more dynamic methods to detect spam accounts.

**Methods**. Fuzzy logic (FL) based approaches, as they are the models such that generate results by interpreting the data obtained based on heuristics viewpoint according to past experiences, they can provide robust and dynamic solutions in spam detection, as in many application areas. For this purpose, a data set was created by collecting data on the twitter platform for spam detection. In the study, fuzzy logic-based classification approaches are suggested for spam detection. In the first stage of the proposed method, a data set with extracted attributes was obtained by applying normalization and crowdsourcing approaches to the raw data obtained from Twitter. In the next stage, as a process of the data preprocessing step, six attributes in the binary form in the data set were subjected to a rating-based transformation and combined with the other real-valued attribute to create a database to be used in spam detection. Classification process inputs were obtained by applying the fisher-score method, one of the commonly used filter-based methods, to the data set obtained in the second stage. In the last stage, the data were classified based on FL based approaches according to the obtained inputs. As FL approaches, four different Mamdani and Sugeno fuzzy inference systems based on interval type-1 and Interval Type-2 were used. Finally, in the classification phase, four different machine learning (ML) approaches including support vector machine (SVM), Bayesian point machine (BPM), logistic regression (LR) and average perceptron (Avr Prc) methods were used to test the effectiveness of these approaches in detecting spam.

**Results**. Experimental results were obtained by applying different FL and ML based approaches on the data set created in the study. As a result of the experiments, the Interval Type-2 Mamdani fuzzy inference system (IT2M-FIS) provided the highest performance with an accuracy of 0.955, a recall of 0.967, an F-score 0.962 and an area

Corresponding author
Oğuzhan Çıtlak,
oguzhan.citlak1@gazi.edu.tr

under the curve (AUC) of 0.971. However, it has been observed that FL-based spam models have a higher performance than ML-based spam models in terms of metrics including accuracy, recall, F-score and AUC values.

## INTRODUCTION

The Internet is widely used around the world, and it can be accessed from almost anywhere, especially with mobile internet devices. With devices such as mobile phones, tablets, laptops that support wireless network connection, users can access the Internet whenever and wherever they want. Social networks such as Twitter, LinkedIn, Instagram, and WhatsApp have become platforms that are widely used by individuals, companies, public institutions, and almost all segments of society with the spread of the internet. These platforms are online channels where people with common interests and purposes can share information and engage in social interactions. This widespread use of social networks has turned them into channels primarily preferred by ill-intended people. Unwanted messages sent to users, malicious links or fake news/ads are among the risks that threaten social network users. Therefore, to minimize the unwanted effects of spam accounts created by ill-intended people targeting social network users, studies carried out on pre-detection, blocking, and taking measures against them are increasing day by day. Twitter, a social networking platform primarily preferred by ill-intended people, sees spam as a problem and uses a number of spam filters to protect its users against spam (*Twitter, 2022a*; *Song, Lee & Kim, 2011*; *Thomas et al., 2011*; *Delany, Buckley & Greene, 2012*; *Rybina, 2012*; *Bouadjenek et al., 2022*). Twitter, which is exposed to new methods and behaviours developed by spam accounts, may be inadequate in detecting these malicious accounts from time to time. Therefore, it has become inevitable to use more dynamic and powerful methods to detect spam accounts.

In the literature, many methods are proposed for detecting spam accounts in the Twitter social network. The anomaly detection method (ADM), link analysis method (LAM), comparison and contracting method (CCM), deceptive information detection approach (DIDA), following and follower comparison analysis (FFCA), ensemble learning analysis (ELA), account creation time-based analysis (ACTA), using spammer detection tools (USDT), honeypot-based twitter spam detection (HTSD), short message analysis (SMA), trend-topic analysis (TTA), tweet-based spam detection (TSD), graph-based spam detection (GSD), and hybrid spam detection (HSD) are among these methods, which are commonly used approaches to detect spam (*Talha & Kara, 2017*; *Çıtlak, Dörterler & Doğru, 2019*; *Güngör, Ayhan Erdem & Doğru, 2020*; *Rupapara et al., 2021*; *Bouadjenek et al., 2022*).

Spam detection with ADM is based on the determination of normal or abnormal behaviour of social network users with sudden behavioral changes, based on their basic characteristics (*Anantharam, Thirunarayan & Sheth, 2012*; *Fernandes, Patel & Marwala, 2015*; *Rahman et al., 2021*). An anomaly is a behavior in data that does not conform to well-defined normal behavior. The uppermost advantage of this method is the possibility of discovering previously unknown spam movements, but the high rate of false alarms in spam detection, that is, the high rate of negative/positive alarms, brings with it a significant disadvantage. Behaviours describing normal activities, can change over time. For this reason, it may not always be possible to define normal behavior. Link shortening is a hyperlink technique that helps reach long or complex URLs or IP addresses over a shorter link. LAM methodologically analyses malicious tweeter links created by shortening URLs to include words typical of current popular news topics (*Benevenuto et al., 2010*; *Wang et al., 2013*; *Daffa, Bamasag & AlMansour, 2018*). A problem with this method is that URLs leading to malicious websites reduce the efficiency of real-time web search services. CCM, also known as contrast comparison, is an effective approach that detects spam by analysing real users and non-real users with ML-based approaches (*Martinez-Romo & Araujo, 2013*; *Clark et al., 2016*; *Adewole et al., 2020*). However, as a result of the method following real account owners for analysis, a problem may arise such as these accounts marked as spam by Twitter. DIDA is an approach that identifies deceptive information and content that directs users to harmful websites and detects accounts that transmit this information (*Alowibdi et al., 2015*; *Chen et al., 2017*). While the frequency of spreading deceptive fake information on social networks is seen as an advantage of this method, it is a disadvantage for normal users to use this information correctly. For example, the social news that "you have won a free mobile phone, click on our website below" is not believable. No one will give anyone a free cell phone. This news should be considered false. The FFCA deals with the "follower" and "friend" relationships between Twitter users. The spam detection strategy of this method is based on comparing the active relationships between spammers and regular users and analysing the results (*Jeong et al., 2016*). FFCA's management of the process through the number of follows and followers affects its performance based on the difference between. A spam account may have a large number of users followed, but not many friends. Although spam accounts are usually created recently, they have a large number of accounts they follow. The effect of the uneven distribution between ELA and spam and non-spam classifications on the spam detection rate is analysed (*Liu et al., 2016*; *Liu et al., 2017*; *Madisetty & Desarkar, 2018*; *Kaddoura et al., 2022*). Ensemble methods use more than one classifier in the same classification task, it is a kind of common sense operation. What is similar with machine learning is that you get better and healthier results with different models compared to single models. This method, which uses community learning, has a high probability of being marked as spam for normal users, too, which is a significant disadvantage. ACBTA is an approach for spam detection that uses analysis of the frequency and diversity of messages posted on the Twitter social network, while also based on the time the account was created (*Chen et al., 2014*; *Eshraqi, Jalali & Moattar, 2015*; *Chaturvedi & Purohit, 2022*). The most important disadvantage of this method is that it produces lower results for text analysis. Finally, spam accounts and messages are

endeavored to be detected through algorithms and external software such as Integro, SybilRank, Pajek, and ReDites in USDT (*Batagelj & Mrvar, 1998*; *Osborne et al., 2014*; *Boshmaf et al., 2015*; *Gao et al., 2015*). Third-party tools used with this method must have high reliability. Such tools can sometimes even be blocked by antivirus software.

When the disadvantages of the existing methods are generally analyzed, in this study, Interval type-2 FL and type-1 FL-based spam detection models based on Mamdani and Sugeno type fuzzy inference approaches have been developed to propose a robust and dynamic solution for the detection of spam in social networks. In order to test the models developed, 4 different FL-based spam detection models, namely IT2M-FIS, IT2S-FIS, T1M-FIS, and T1S-FIS, were applied to a dataset being adapted to inputs of the models by first feature extraction process, and then data reconciliation process to the raw dataset obtained from the Twitter platform. The strength of the models proposed has been verified by testing ML-based spam detection models that include methods of the SVM, BPM, LR, and Avr Prc built in the Azure ML studio, on the same dataset. In order to provide an advantage against the studies for spam detection, the datasets used in the fuzzy logic-based method we recommend were also evaluated with machine learning-based methods and the results were interpreted. At the same time, the results were compared with the results of similar studies in the literature. Contributions of the proposed study may be summarized as (a) presenting FL-based approaches based on IT2M- FIS and IT2S-FIS as a new model for spam detection; (b) giving the strong performance of interval type-2 FL based models thanks to their ability to eliminate uncertainties arising from expert experience or environmental factors with type-2 fuzzy sets they use inside; (c) being faster models of the proposed methods as a result of not having a learning phase and having less computational load than some ML and deep learning-based methods, and giving a competitive performance with these methods; (d) to be able to produce high performance if they are designed with a good expert knowledge experience; and (e) reflecting the effect of most features on the output performance as a result of combining binary-valued features.

The remaining parts of the study are organized as follows: In the second section, the studies including artificial intelligence approaches based on FL, deep learning, and ML used in the detection of spam in the Twitter social network are presented in a summarized manner together with their results. In the third section, the materials and methods used for the spam detection processes in the Twitter social network and the performance metrics used to measure the performance of the methods applied are explained in detail. In the fourth section, the experimental results obtained for all models are presented, and their results are discussed. In the last section, general evaluations of the results are given.

## RELATED STUDIES IN THE LITERATURE

Numerous studies have been conducted in the literature on spam analysis and detection. Under the title of this chapter, studies using artificial intelligence methods including ML algorithms, FL algorithms, and deep learning algorithms related to the study we present herein will be discussed.

*Ouni, Fkih & Omri (2022)* detected spam tweets through a new approach proposed based on the extraction of new topic-based features (TOBEAT) from Twitter data, which is also based on CNN (convolutional neural network), and BERT (bidirectional encoder representations of transformers). Experimental studies have shown that CNN architecture is a suitable classifier for spam detection. The model they proposed exhibited a very high performance with 0.9497 accuracy, 0.9405 precision, 0.9588 recall, and 0.9495 F-measurement values.

Malicious expressions, mocking expressions, mobbing expressions, and hate speech can be considered spam on the Twitter social networking platform. *Ayo et al. (2021)* developed a model with a probabilistic clustering method to overcome the classification problems related to hate speech on the Twitter platform. They labelled the tweets they obtained using metadata extractors in two categories as containing hate speech and not containing hate speech, through crowdsourcing experts. They developed the features they represented with the model term frequency-inverse document frequency (TF-IDF) through the subjects extracted over the Bayesian classifier. They then classified hate speech by applying the FL approach to this data. With the model in which 5-fold cross-validation was applied, they achieved results of 0.9453 accuracy, 0.9254 precision, 0.9256 F-score, 0.9174 recall, and 0.9645 AUC in performance.

In a study conducted by *Meriem, Hlaoua & Romdhane (2021)*, a fuzzy mock detection approach was proposed using social information such as past tweets, replies, and likes, which were taken into account by their severity. In their study, the Twitter datasets obtained by SemEval2014 (*Rosenthal et al., 2014*) and *Bamman & Smith (2015)* were used. The evaluation results showed that the use of FL made significant contributions to the improvement of the precision and accuracy metrics of the classification. It showed that using severity yielded better results in terms of recall, precision, and accuracy metrics than existing approaches. From the test results they obtained with different combinations of the severity of each feature, they reached an average of 0.9573 precision, 0.8737 F-score, and 0.8169 recall values.

*Liu et al. (2017)* addressed the problem of class imbalance in Twitter spam detection in their study, using community learning. In this study, it was argued that the uneven distribution between the spam and non-spam classes on Twitter had a large impact on the spam detection rate. Also, a fuzzy-based model over-sampling method (FOS) that generates synthetic data samples from observed samples based on fuzzy-based information was recommended. The Twitter dataset (*Liu et al., 2016*), which was used in another study of theirs before, was also used in this study. They applied random over-sampling (ROS) for spam classes in the Twitter dataset, and random under-sampling (RUS) methods for non-spam classes. They compared the models they developed with each other by running the naive Bayesian classifier (NB), K-nearest neighbours (KNN), SVM, RUBoost, C4.5 decision tree algorithms, and Ensemble learning models on Weka. The highest average (Avg.) precision was shown as >78%. In addition, some results obtained in detecting spam messages areas follow $FP_{FOS}$ 9.6%, $FP_{ROS}$ 10.6%, $FP_{RUS}$ 16%, Avg. F- Measure$_{FOS}$ 61%, Avg. F- Measure$_{ROS}$ 60%, Avg. F- Measure$_{RUS}$ 57%, Avg. precision >78%, Ensemble learning precision 82%–90%, and true positive rate (TPR) TPR 75%.

*Gupta et al. (2018)* proposed a Hierarchical Meta-Path (HMPS)-based model to detect spammers using their phone numbers to promote campaigns on Twitter. For the model they proposed, information about 3,370 campaigns was collected through the metadata of 670,251 users on Twitter, and a heterogeneous network model was created by using the interaction between different nodes in the dataset created based on the information obtained. Experimental results were obtained from online social networks (OSN) with different features, which were configured through this model. In order to compare the performance of the proposed model, logistic regression (LR), latent dirichlet allocation (LDA), K-nearest neighbours (KNN), decision tree (DT), NB, random forest (RF), and SVM algorithms were used on active learning framework platform. Performance evaluation was carried out over standard information retrieval metrics, namely precision, recall, F1-score, and area under the ROC curve (AUC), and the values obtained were 0.95, 0.90, 0.93, 0.92, respectively for the recommended method HMPS + OSN2.

In a study carried out by *Ameen & Kaya (2018)*, a model was developed for spam detection in social networks using the deep learning method. The Word2Vec tool was used because it is a time-consuming and tedious manual task to check the URL messages in the Twitter dataset used in the study and to extract their features. Then binary classification methods were used to separate spam from non-spam tweets. A deep learning model known as multi-layer perceptron (MLP) has been applied for the spam classification task. RF, decision tree (J48), and NB methods were also used for comparison classification. When they compared the MLP method with Word2Vec to other methods, they realised that it performed much higher. They obtained 0.92 precision, 0.88 recall, and 0.89 F-score performance values in their proposed method.

*Madisetty & Desarkar (2018)* proposed a community model combining five CNN-based and one feature-based method for spam detection on Twitter. CNN was trained with word insertions of different sizes using the Glove and Word2Vec tools. In their studies, they utilised word embedding features in deep learning methods, and user-based, content and N-gram features in the feature-based method. They used two different datasets, one balanced (HSpam) and the other unbalanced (1KS10KN). Experimental results of the proposed model in different datasets were obtained as $\text{Precision}_{1KS10KN}/\text{Precision}_{HSpam14}$ (0.922/0.880), $\text{Recall}_{1KS10KN}/\text{Recall}_{HSpam14}$ (0.867/0.909), and $\text{F-score}_{1KS10KN}/\text{F-score}_{HSpam14}$ (0.893/0.894). In terms of accuracy and AUC values, 0.957 and 0.9643 were obtained, respectively.

*Ashour, Salama & El-Kharashi (2018)* endeavored, in their study, to detect spam tweets using N-gram character features. In the study, the performances of different N-gram feature representations (TF, TF-IDF) were evaluated with supervised learning classifiers LR, SVM, and RF. Social Twitter Honeypot dataset hosting labelled versions of spam, and non- spam data was used as a dataset. The results of the study proved that the linear classifiers SVM and LR had better performance than the tree-based classifier RF. A recall value of 0.794, a precision value of 0.795, and an F-score value of 0.794 were obtained with the LR classifier, in which N-gram character features have TF-IDF.

# MATERIALS & METHODS

In this section, it is first presented how to extract the raw dataset used in our study from the Twitter platform. Then our proposed methodology for spam detection in social networks based on the processing phases is explained. The feature extraction and data reconciliation procedures in the data processing phase, and the "Feature Selection" and "Classification" procedures in the data evaluation phase, together with the methods being used are given in detail.

## About dataset

Datasets are used in most of the studies on social networks. Twitter, one of the most widely used platforms of these networks, provides an Application Programming Interface (API) for software developers to use in their works. The software developers draw Twitter data from an account *via* this interface through a program written in a programming language. The data extraction process *via* API requires the knowledge of account-specific keys such as Consumer Key, Consumer Secret, Access Token, and Access Token Secret on the screen of the interface shown in Fig. 1A (*Twitter, 2022b*). As the platform on which the program to be used in extracting Twitter data will be developed; Python has a very significant position because it does not need a compiler, the code-writing process is fast, and it contains the libraries needed for Twitter application developers (*Van Rossum, 2007*). Therefore, in this study, Python was used as the program development platform. The raw dataset was created by drawing the data of 1,225 accounts from the API through the program written in Python using the keys in Fig. 1A, which belongs to the @oguzhancitlak account. Twitter may suspend abusive accounts based on its spam policy (*Twitter, 2022a*). Figure 1B shows the status of such an account. However, Twitter may be inadequate in detecting these malicious accounts from time to time.

In Fig. 1C, a screenshot of the data in Json format obtained from Twitter is presented. Json (Javascript object notation) is a data format in which raw files taken over Twitter are saved. At the same time, Twitter uses this file format to process the raw data it has on it.

## Proposed methodology for spam detection in social networks

The scheme of the research methodology proposed for spam detection in social networks is shown in Fig. 2. With the proposed scheme, it is aimed to provide a fast, dynamic, and effective solution to the detection of spam in social networks through FL-based models, based on the intuitive view of people in solving a problem and making inferences according to their past experiences. Models including Mamdani and Sugeno type fuzzy inference systems based on T1- FL and IT2-FL approaches were used for spam detection. Related methodology runs two main phases, namely "Data Processing" and "Data Evaluation" for spam detection.

The data processing phase includes feature extraction and data reconciliation procedures. The feature extraction procedure allows the removal of terms that do not make any sense from the raw dataset in Json format, which is expressed as the Twitter dataset, and the extraction of features using the crowdsourcing method. The data reconciliation procedure processes data in a form that cannot be processed by FL-based models, transforms it into

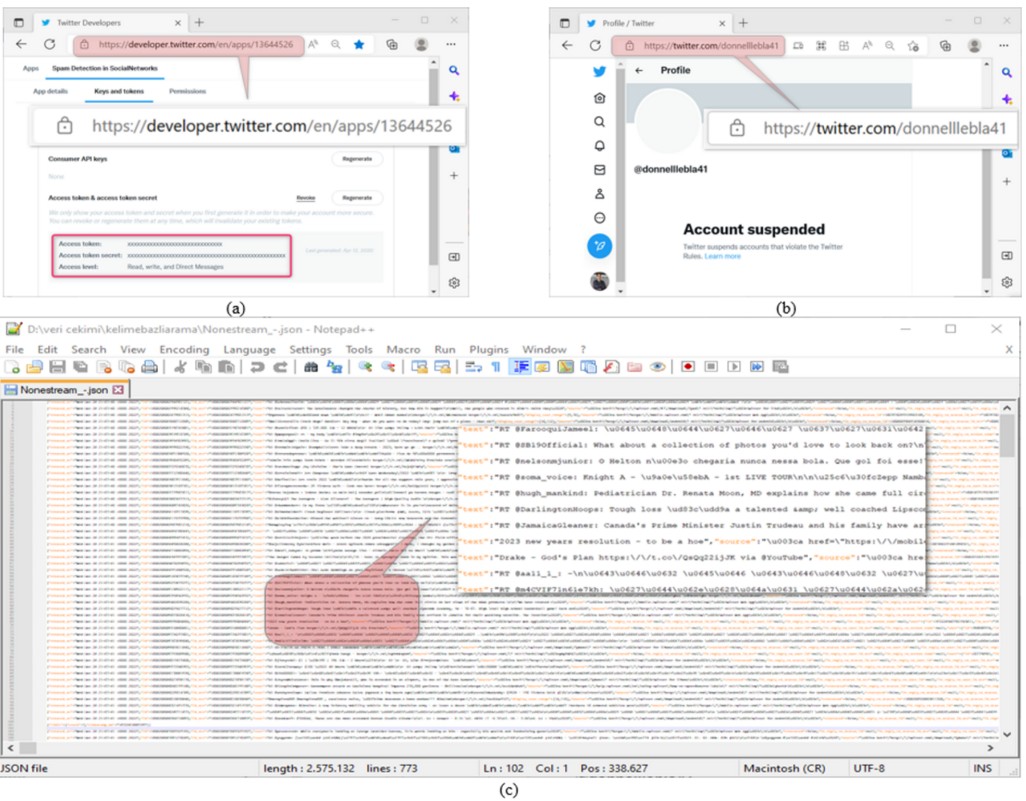

**Figure 1** (A) Twitter application developer API platform, (B) an account suspended by Twitter, (C) a section from the resulting Twitter dataset.

data that can be applied to classifier inputs and creates databases in CSV and XLSX formats with these data. In this process, digitization and multiplexing of the data in accordance with the criteria range, dividing the data into binary and real-valued data, converting binary data into real data, and recombining the data into databases in CSV and XLSX data formats is performed in an orderly fashion. The data evaluation phase consists of feature selection and classification processes. This phase determines the classifier entries by applying the Fisher score feature selection method to the real-valued data converted and transferred to the database and based on these entries, it decides whether the data coming through the classifiers is spam.

### Data processing phase

The data processing phase includes the processes that are realised from extracting the characteristics of the raw Twitter data to make this data suitable for the inputs of the models used as classifiers. The processing steps used to carry out these processes are presented under sub-headings below.

*Feature extraction:* The calculations in the raw dataset obtained through the Twitter API with the program Python bring many features along with them. In the process before deciding which of these features will be decisive in determining whether the accounts

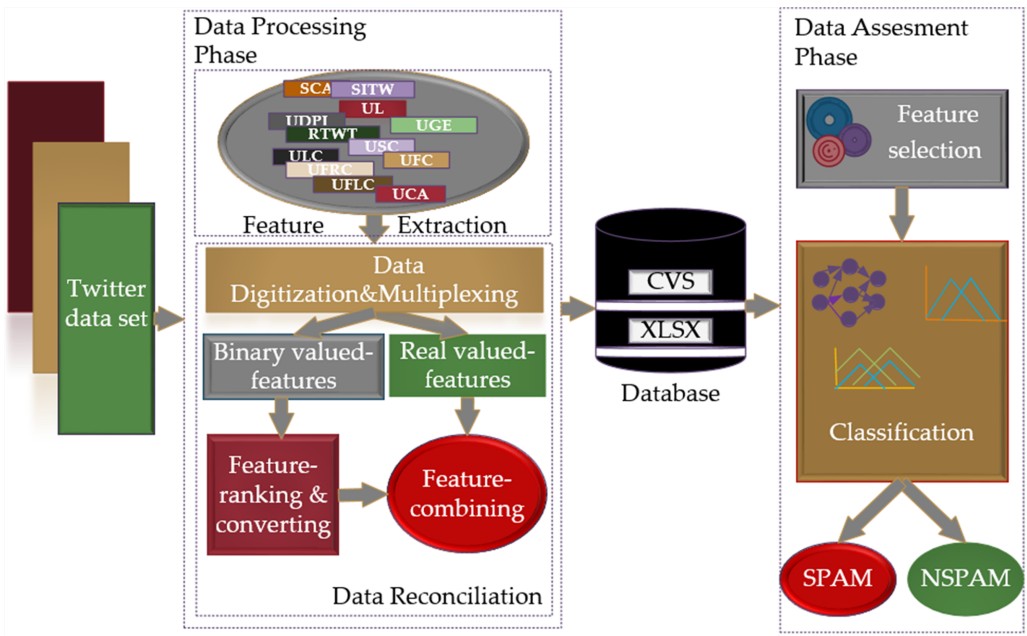

**Figure 2** The proposed research methodology for the spam detection in social networks.

are spam or not and whether they will be included in the feature inference set, words, terms, or characters that do not make any sense in the dataset are extracted. This is called normalizing the dataset. Normalization helps to deal with "detecting out-of-vocabulary (OOV) words" for an infinite number of possible expression combinations and to convert various expressions representing similar emotions into a singular form (*Arslan & Orhan, 2019*). With this method, terms that are not part of normal words in the natural language processing environment are cleaned from the dataset. Thus, a more meaningful and more accurate raw data processing is obtained. Extracting the features from the dataset obtained after the normalization process and determining the real class (Identification of spam (Yes) or non-spam (No) accounts) is carried out manually based on expert opinion. Crowdsourcing is used methodologically in the implementation of the process. The basis of this method is based on the coming together of people in the communication network on the internet, independently examining the project for which support is requested, and drawing a common conclusion in order to realize a particular project (*Brabham, 2008*; *Bücheler & Sieg, 2011*). In Table 1, the features obtained as a result of this common result, their criteria ranges, and explanations are shown.

As seen from the Table 1, a dataset consisting of 1,225 spam data with 12 features is obtained.

*Data reconciliation:* At the end of the feature extraction process, some of the features were labelled as binary-valued features such as "Yes-No" and "True-False" while others were labelled with a certain criterion confidence interval such as "0-99" and "900-999". It may be possible to detect spam by applying some ML-based models to data with these features,

**Table 1  Taxonomy and criteria ranges of the dataset used.**

| No | Type in model | Type | Evaluation range | Explanation |
|---|---|---|---|---|
| 1 | User_Statuses_Count (USC) | Tweets | 0-99,100-199, …,1000000-1999999 | If possible, it shows that the user's most recent Tweets or retweets. In some circumstances, this data cannot be provided, and this field will be omitted, null, or empty. Perspectival attributes within Tweets embedded within users cannot always be relied upon. |
| 2 | Sensitive_Content_Alert (SCA) | Array of object | TRUE/FALSE | It represents sensitive objects included in the text of a Tweet or within textual fields of a user object |
| 3 | User_Favourites_Count (UFC) | Boolean | 0-9,10-19,20-29,…,100000-1999999 | Perspectival indicates whether this Tweet has been liked by the authenticating user. |
| 4 | User_Listed_Count (ULC) | Int | 0-9,10-19,20-29,…,900-999 | It shows the number of public lists that this user is a member of. |
| 5 | Source_in_Twitter (SITW) | String | YES / NO | It shows the Utility used to post the Tweet, as an HTML-formatted string. Tweets from the Twitter website have a source value of web. |
| 6 | User_Friends_Counts (UFRC) | Int | 0-9,10-19,20-29,…,1000-99999 | The number of users this account is following (AKA their "followings"). Under certain conditions of duress, this field will temporarily indicate "0" |
| 7 | User_Followers_Count (UFLC) | Int | 0-9,10-19,20-29,…,100000-1999999 | The number of Tweets this user has liked in the account's lifetime. |
| 8 | User_Location (UL) | String | YES / NO | It shows the the user-defined location for this account's profile. Not necessarily a location, nor machine-parseable. This field will occasionally be fuzzily interpreted by the Search service. |
| 9 | User_Geo_Enabled (UGE) | Boolean | TRUE/FALSE | Perspectival indicates whether this Tweet has been liked by the authenticating user. |
| 10 | User_Default_Profile_Image (UDPI) | Boolean | TRUE/FALSE | When true, it indicates that the user has not uploaded their own profile image, and a default image is used instead. |
| 11 | ReTweet (RTWT) | Boolean | TRUE/FALSE | It indicates whether this Tweet has been retweeted by the authenticating user. |
| 12 | Account Suspender (CLASS) | Boolean | TRUE/FALSE | It shows the account taken as spam or not Fig. 1B |

which are not in numerical form. However, in order to work with FL-based models, its inputs must be real values. Data reconciliation is a procedure that can be used both to fulfill these purposes and to increase the performance of all models in this process. Digitising and multiplexing data, dividing data with binary and real-valued features, grading binary features and converting them into real-valued features, and recombining them are the operations performed by this procedure.

*Digitisation and multiplexing of data:*  In this process, the values of the binary-valued features labelled with the feature value "Yes-No" and "True-False" in the dataset with extracted features are replaced by the values "1" and "0" (1 = Yes, True; 0 = No, False),

and for the data with the characteristics labelled with the criteria interval, five different real values were randomly generated for each data in the relevant criteria confidence interval. In this case, the binary feature values in the data corresponding to the criterion confidence interval were also multiplied 5 times as the same value. Data of 1,225 pieces turned into 6,125 pieces of data at the end of this process.

*Splitting multiplexed data:* The splitting process is the splitting of the data in the multiplexed dataset into two separate matrices, separating the binary and real value features. At the end of this process, the multiplexed data is divided into two different datasets, binary-valued features and real-valued features. While the set of binary-valued features contains 6,125 data with six binary-valued features, the set of real-valued features consists of a total of 6,125 data, 1 of which represents the binary-valued real class (RC), and five of which has six real-valued features.

*Feature ranking and converting:* Converting the values of binary-valued features to real-number values covers a process based on the conversion of data with six columns and 6,125 rows consisting of "1" and "0" to the decimal number system after being represented in binary number form according to the importance of features. In the first stage of this process, which takes place in two stages, the feature selection algorithm is applied to the binary-valued data obtained at the end of the division process to determine the digit name and rank based on the importance degree of the features. It is used to determine at which step the features to be represented in the form of binary numbers will take place. In the feature ranking process, the Qui-Square method, which is common among statistical methods, was used. The test formula for this method, which is used to measure the relationship between categorical variables, is given in Eq. (1).

$$X^2 = \sum_{i=1}^{n} \frac{(O_i - e_i)^2}{e_i} \tag{1}$$

where $n$ represents the number of the features in the dataset, $O_i$ the observed frequency value for the i'th feature, and $e_i$ the expected frequency value for the i'th feature (*Uzun & Ballı, 2022*). The sorting of the features is carried out in descending order of $X^2$ value. In Fig. 3, the test gains ($X^2$) of each feature in the dataset according to the chi-squared method and the digit names of the features in the binary number system based on these values are shown.

According to the gain ordering, the sequence consisting of the binary-valued features in the form of "SCA RIWT SITW UDPI UGE UL" from the highest gain to the lowest gain will give the binary number equivalent of these features. In the second stage of the process, the real value of this number can be found by obtaining its decimal equivalent as seen in the equation below.

$$CBFTR_i = 32 \times SCA_i + 16 \times RTWT_i + 8 \times SITW_i + 4 \times UDPI_i + 2 \times UGE_i + 1 \times UL_i \tag{2}$$

where, $i$ is the row index of the data. By using Eq. (2), combined real values of 6,125 binary features are obtained.

| FTR. Name | Sensitive Content Alert (SCA) | Re-Tweet (RTWT) | Source in Twitter (SITW) | User Default Profile Image (UDPI) | User Geo Enabled (UGE) | User Location (UL) |
|---|---|---|---|---|---|---|
| FTR_Inf. Gain | 0,3532 | 0,0435 | 0,0329 | 0,0058 | 0,0048 | 0,0038 |
| | $2^5$ | $2^4$ | $2^3$ | $2^2$ | $2^1$ | $2^0$ |
| Digit Name | 32 | 16 | 8 | 4 | 2 | 1 |

**Figure 3** Test gains ($X^2$) and digit names of binary-valued properties obtained with the chi-squared method.

*Feature combining:* Feature combining process combines the 6125×1 vector that gives the real values (CBFTR) of the binary-valued features obtained at the end of the feature ranking and converting process with the 6125×6 matrix containing 6 features (USC, UFC, ULC, UFRC, UFLC, RC), with five real-valued and one binary-valued RC data obtained in the process of dividing the data and obtains a database consisting of 6125 spam data with seven real-valued features in CSV and XLSX format. In Fig. 4, the preview of the database containing the final spam data, which has been prepared for the evaluation process through the processes in the data reconciliation procedure, is presented.

The real values of the CBFTR feature in the first column of the database define the combined state of the 6 binary-valued features. The data in the last column represents the RC and carries information about whether the data in the row to which it belongs to is spam or not. An RC value of "1" indicates that the data in the relevant row is spam, and a value of "0" indicates that the data is not spam.

### Data assessment phase

The procedure for the selection of the feature to be implemented in the data evaluation phase and the presentations of ML and FL-based spam detection models as well as the performance metrics to be used in the evaluation are explained within the scope of this section.

*Feature selection:* In this study, the feature selection procedure was used to create spam detection models with less input, simpler in structure, easy to interpret and high performance by eliminating unnecessary features. The process can shorten the training time and minimize overfitting problems in ML-based models. This situation affects the performance of the models used as classifiers positively. Enforcement of this process in FL-based models is even more important than it is in ML-based models. Because in these models, the fuzzy universes to be defined for the inputs and outputs, the fuzzy sets in these spaces, and the rules determining the relationship between input and output are decided according to expert knowledge and experience, the reduction of the input number and the structural simplification of the models increase the interpretability of these parameters and computationally, it allows building models that are accurate for spam detection. Therefore, it is very important to use an effective feature selection method in the feature selection processes of both ML and FL-based spam detection models. In this study, the feature

| | A | B | C | D | E | F | G |
|---|---|---|---|---|---|---|---|
| 1 | Combined | User | User | User | User | User | |
| 2 | Binary | Statuses | Favourites | Listed | Friends | Followers | Real |
| 3 | Feature | Count | Count | Count | Counts | Count | Class |
| 4 | (CBFTR) | (USC) | (UFC) | (ULC) | (UFRC) | (UFLC) | (RC) |
| 5 | 60 | 781 | 1097 | 1 | 514 | 865 | 0 |
| 6 | 60 | 790 | 1278 | 9 | 542 | 803 | 0 |
| 7 | 60 | 712 | 1546 | 9 | 591 | 884 | 0 |
| 8 | 60 | 791 | 1957 | 4 | 579 | 893 | 0 |
| 9 | 60 | 763 | 1964 | 8 | 595 | 867 | 0 |
| 10 | 27 | 27060 | 28234 | 743 | 248 | 32760 | 0 |
| 11 | 27 | 26048 | 26048 | 738 | 244 | 36797 | 0 |
| 12 | | | | | | | |
| 6122 | | | | | | | |
| 6123 | 12 | 2774 | 9 | 6 | 7 | 14 | 1 |
| 6124 | 12 | 2435 | 7 | 1 | 0 | 15 | 1 |
| 6125 | 15 | 1122 | 7 | 3 | 69 | 36 | 1 |
| 6126 | 15 | 1299 | 6 | 2 | 69 | 37 | 1 |
| 6127 | 15 | 1136 | 9 | 9 | 66 | 37 | 1 |
| 6128 | 15 | 1052 | 5 | 1 | 61 | 30 | 1 |
| 6129 | 15 | 1384 | 6 | 5 | 67 | 35 | 1 |

**Figure 4   A cross-section of the spam data obtained at the end of the data reconciliation processes in the XLSX database.**

selection process was implemented using the Fisher score method, which is one of the frequently used filter-based feature selection methods. This method produces a score by using the mean and standard deviation values of the features for each column, as shown in Eq. (3), and ranks the features based on the score it produces from high to low to form the feature set suitable for the criterion (*Ferreira & Figueiredo, 2012*).

$$fs(x_i) = \frac{|\mu_i^+ - \mu_i^-|}{\sigma_i^+ - \sigma_i^-}. \tag{3}$$

The "+" and "−" given in the equation represent two different classes, positive and negative. $\mu_i$ represents the mean of each class, and $\sigma_i$ indicates its standard deviation. The Fisher score method calculates a correlation score for each class using the mean and standard deviation values of the features. A high Fisher score indicates that the mean difference between the two classes for the relevant trait is large and there are small deviations in its value in the related classes (*Budak, 2018*). By applying the Fisher score feature selection method to the dataset obtained as a result of the feature combining process, 1.523, 0.0778, 0.0214, 0.0211, 0.00179, and 0.000127 gain scores were obtained

for the CBFTR, USC, UFC, ULC, UFRC, and UFLC features, respectively. It is seen that the combined binary-valued features shown with CBFTR have a very high score when compared to other features. This is an indication that the binary-valued features obtained by sorting and transforming according to the decimal value of the binary number form will contribute to the classification process. USC is the feature with the 2nd highest Fisher score of the message count. It is understood that this feature has a high score when compared to other features. The UFC is a feature that displays the number of favourites added. Its score is lower when compared to the two selected features. These three features were selected and applied to both ML-based models and FL-based models. We chose to use the Fisher score or Benforroni mean (BM) model in this calculation. BM operator is an important and meaningful concept to examine the interrelationships between the different attributes. BM method gives very successful results in many studies (*Liu et al., 2020*). However, the structure of the dataset used is unsuitable.

*Classification with FL-based models:* The FL-based models used in this study are calculators that find out the degree to which the data obtained as a result of the feature selection process is spam. The result obtained from the output of these models does not give us whether the incoming data is directly spam or not. The classified result (SPAM or NON-SPAM) can be obtained the by passing the FL-based model output through the evaluation function. In Eq. (4), the function used to evaluate the outputs of FL-based models is presented.

$$EV_{FL}(DOSPAM(CBFTR, USC, UFC)) = \begin{cases} 1 & if\ DOSPAM(CBFTR, USC, UFC) \geq 0.5 \\ 0 & else \end{cases} \quad (4)$$

where CBFTR, USC, and UFC show the inputs of the FL-based model, the DOSPAM defines FL-based model's output and the $EV_{FL}$ represents the output of the evaluation function. The output of the evaluation function returns the status of whether there is spam or not, and its value is equal to 1 or 0 (1: spam and 0: not spam). In practice, it is possible to come across different types of fuzzy inference methods such as Mamdani inference system, Sugeno inference system, Tsukamoto inference system, and Larsen inference system as FL-based systems. However, the fact that the Mamdani inference system appeals to more human perception and has more interpretability, and that the Sugeno inference system easily obtains the defuzzified result has led to the widespread use of both fuzzy inference systems in applications (*Khosravanian et al., 2016*). In our study, four different spam detection models including Mamdani and Sugeno type fuzzy inference systems based on T1-FL and IT2-FL approaches were used: T1M-FIS, T1S-FIS, IT2M-FIS, and IT2S-FIS. Models were built in the MATLAB platform. The most important difference between Mamdani-type fuzzy inference systems and Sugeno-type fuzzy systems is manifested in rule outputs. While, in a Mamdani-type fuzzy inference system, the rule output is represented by a fuzzy set, in a Sugeno-type fuzzy inference system it is represented by a function. Therefore, the defuzzification process is performed more easily in Sugeno-type fuzzy inference systems. Apart from that, all processes are the same in both systems. Accordingly, here, fuzzy logic-based approaches are explained over Mamdani-type fuzzy inference systems. Only different parameters are explained for Sugeno-type fuzzy inference systems. The CBFTR,

USC, and UFC features obtained at the end of the feature selection process were assigned as the input variables of the FL-based models. As the model output, the degree of spam (DOSPAM) variable, which its score varies between 0 and 1, was used.

*T1-FL approach:* T1-FL approach, known also as the traditional FL system corresponds to the widely used Mamdani type fuzzy inference system. In Fig. 5, the block diagram of T1M-FIS with three inputs and one output built in the MATLAB platform is shown to calculate the spam degree of data coming from the spam database *via* the feature selector.

T1M-FIS, which consists of four basic units: fuzzifier, rule base, inference engine and defuzzifier, verbally labels the real values of three features that come to its inputs during the fuzzification process, by converting them into appropriate linguistic values. The management of the process is carried out through fuzzy sets represented by membership functions defined separately for each input within the fuzzifier unit. Determining the boundaries of the input fuzzy universes, and the fuzzy set types and fuzzy set numbers in the fuzzy universes are the basic criteria to be used in the configuration of the fuzzifier. In this context, the boundary values of fuzzy universes for CBFTR, UFC, and USC inputs were obtained as "0-70", "0-80000" and "0-1000000", respectively. When the input variable columns in the database were compared with the real class column, it was seen that when all input variables had a low score (this corresponds to approximately 15% of the highest score in each input column) incoming data was labelled as spam, otherwise not as spam. Therefore, for each input variable, it was sufficient to use 3 fuzzy sets, represented by the linguistic score value low (L), medium (M), and high (H) labels. Since CBFTR is a real-valued input obtained as a result of combining six binary-valued features by ranking based on the chi-square test gain, it contains a non-linear structure. Many data sets can be modeled with the Gaussian function. Therefore, it is quite common to assume that the clusters in the datasets come from different Gaussian function. In other words, Guassion is the model described as a mixture of k-piece Gaussian Distributions, under the assumption of normality in the data set. This is the main idea of this model (*Shuster, 1968*). Therefore, this mathematical input equation is represented by fuzzy sets defined by the Gaussian combination membership degree function (gauss2mf) given in Eq. (5).

$$
\mu_{X\,CBFTR}(x; \sigma 1, c1, \sigma 2, c2)
$$

$$
= \begin{cases} \mu_{Gauss}(x, \sigma 1, c1), & if & x < c1 \leq c2 \\ \mu_{Gauss}(x, \sigma 2, c2), & if & x > c2, c1 \leq c2 \\ 1, & if & c1 \leq x \leq c2 \ and \ c1 \leq c2 \\ \mu_{Gauss}(x, \sigma 1, c1).\mu_{Gauss}(x, \sigma 2, c2), & if & c1 > c2 \end{cases}
$$

$$
\mu_{Gauss}(x, \sigma, c) = e^{\frac{-(x-c)^2}{2\sigma^2}} \tag{5}
$$

where $X$ represents the symbol for type-1 fuzzy sets labelled *L*, *M*, and *H*. Since the follower number and tweet number of an account often changes in direct proportion to the content of that account and the status of its profile, the fuzzy sets for the USC and UFC inputs that give the changes in these numbers are defined by the trapezoidal membership function,

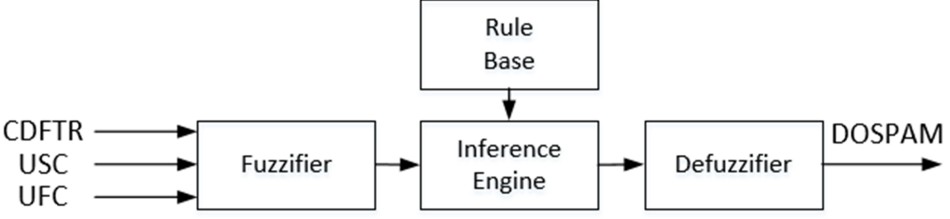

**Figure 5**   **Block diagram of the T1M-FIS proposed for calculating the degree of spam.**

whose formula is given in Eq. (6).

$$
\mu_{X\atop{USC\atop OR\atop UFC}}(x; a, b, c, d) = \begin{cases} \dfrac{x-a}{b-a}, & if & a \le x < b \\ 1, & if & b \le x < c \\ \dfrac{d-x}{d-c}, & if & c \le x \le d \end{cases} \tag{6}
$$

The output of T1M-FIS is defined by a trapezoidal membership function in the fuzzy universe in the range of "0-1" and represented by two fuzzy sets labelled with verbal expressions "NON-SPAM" and "SPAM". For T1S -FIS, these sets are in singleton form, and the constants are determined as 0.283 for NON-SPAM and 0.709 for SPAM. Figure 6 shows the membership functions defined for the inputs and output of T1M-FIS.

Each input of T1M-FIS is represented by three fuzzy sets. Therefore, the number of rules defining the verbal relationship between inputs and output is 27. In the proposed models, these rules were created by examining the changes of values given for the inputs and the changes of values given for the actual class label in the columns of the database. As a result of the examinations, it has been seen that low-scoring changes of all inputs carry the output to "spam" class, while medium or high-scoring changes of one or more inputs carry the output to "not spam" class. A cross-section of the rule table obtained under these conditions is given in Table 2.

Fuzzy inference is a process that is implemented in two stages: implication and aggregation. While the weight of the active rules is determined in the implication stage, the weighted rules are combined in the aggregation stage (*Farid & Riaz, 2022*), and then the inference result is obtained. In T1M -FIS, the implication stage of this process is performed using the "min" operator, and the aggregation stage is carried out using the "max" operator (*Hamid, Riaz & Naeem, 2022*). In T1S -FIS, on the other hand, the "prod" operator is used in the implication stage of the related process, and the "sum" operator is used in the aggregation stage. Finally, in the defuzzification process, while the Centroid method, the formula of which is given in eq7, is used as a method in the T1M-FIS, this function is fulfilled with the Wtaver method in the T1S-FIS.

$$
DOSPAM = \frac{\sum_{i=1}^{n} \mu(x_i) . x_i}{\sum_{i=1}^{n} \mu(x_i)} \tag{7}
$$

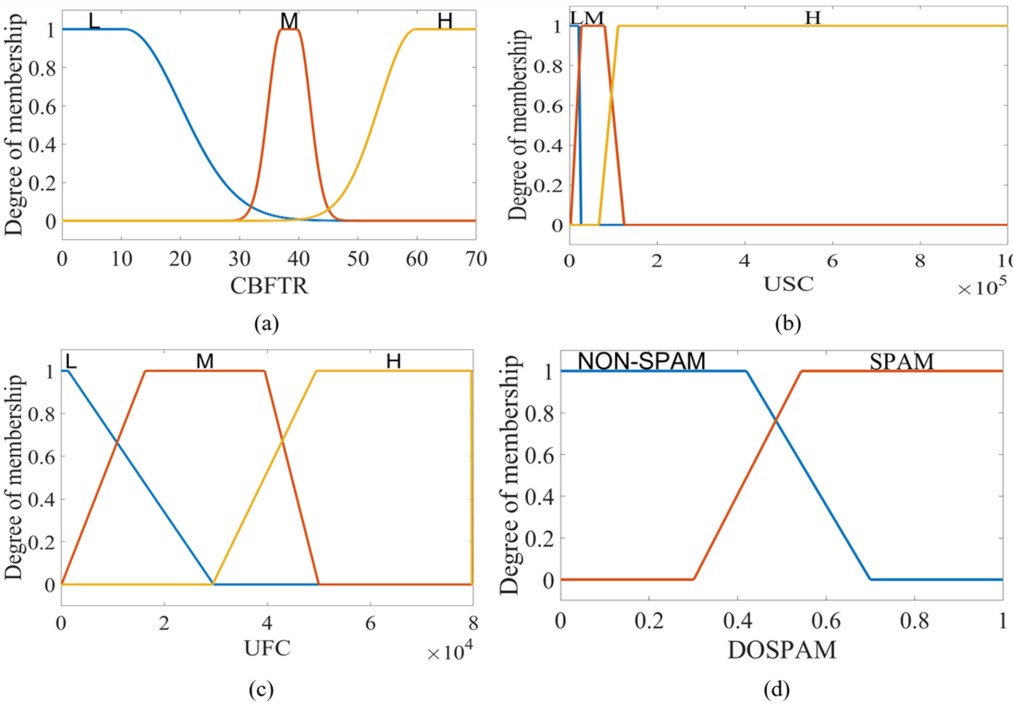

**Figure 6** The membership functions defined for the inputs and outputs of the T1M-FIS: (A) CBFTR input, (B) USC input, (C) UFC input, and (D) DOSPAM output.

**Table 2  A cross-section of the MT-1FL model's rule base.**

| CBFTR input | USC input | UFC input | DOSPAM output |
|---|---|---|---|
| Low (L) | Low (L) | Low (L) | SPAM |
| Low (L) | Low (L) | Medium (M) | NON-SPAM |
| … | … | … | … |
| High (H) | High (H) | High (H) | NON-SPAM |

*IT2-FL approach:* Literature studies and real-time applications have shown that IT2-FL based systems are more effective than T1-FL based systems in the cases where uncertainties prevail (*Atacak & Bay, 2012*; *Ashraf, Akram & Sarwar, 2014a*). As can be seen from the block diagram given in Fig. 7, the structure of IT2M-FIS is quite similar to T1M-FIS except for the type reduction unit. This unit converts the fuzzy inference result in the form of interval type-2 into type-1 fuzzy form in order to apply the defuzzification process to the interval type-2 fuzzy inference result. Although the other units are functionally and structurally the similar as T1M -FIS, Interval type-2 fuzzy sets are used instead of type-1 fuzzy sets in the representation of verbal variables within these units.

Interval type-2 fuzzy sets are 3-dimensional sets that represented by the area called the Footprint of Uncertainty (FOU) between the lower membership function ($\underline{\mu}\tilde{X}$) and its upper membership function ($\overline{\mu}\tilde{X}$). Since these sets have a large number of type-1 fuzzy sets

within this area, they have the potential to capture much more uncertainty than those sets do. Therefore, systems using type-2 fuzzy systems can demonstrate higher performance than systems using type-1 fuzzy systems, especially in problems where uncertainties are dominant. Fuzzifier is used for the incoming values to complete data processing in the degree of spam (*Ashraf, Akram & Sarwar, 2014b*; *Habib, Akram & Ashraf, 2017*). In Eq. (8), the equation of the membership function of the Interval type-2 fuzzy sets defined for the CBFTR input is given.

$$\mu_{\tilde{X}CBFTR}(x) = \left\{ \underline{\mu}_{\tilde{X}CBFTR}(x;\underline{\sigma}1,c1,\underline{\sigma}2,c2), \overline{\mu}_{\tilde{X}CBFTR}(x;\overline{\sigma}1,c3,\overline{\sigma}2,c4) \right\}$$

$$\underline{\mu}_{\tilde{X}CBFTR}(x;\underline{\sigma}1,c1,\underline{\sigma}2,c2) = \begin{cases} \underline{\mu}_{Gauss}(x,\underline{\sigma}1,c1), if & x<c1\leq c2 \\ \underline{\mu}_{Gauss}(x,\underline{\sigma}2,c2), if & x>c2,c1\leq c2 \\ 1, if & c1\leq x\leq c2\, and\, c1\leq c2 \\ \underline{\mu}_{Gauss}(x,\underline{\sigma}1,c1)\cdot\underline{\mu}_{Gauss}(x,\underline{\sigma}2,c2), if & c1>c2 \end{cases}$$

$$\underline{\mu}_{Gauss}(x,\underline{\sigma},c) = e^{\frac{-(x-c)^2}{2\underline{\sigma}^2}}$$

$$\overline{\mu}_{\tilde{X}CBFTR}(x;\overline{\sigma}1,c3,\overline{\sigma}2,c4)$$

$$= \begin{cases} \overline{\mu}_{Gauss}(x,\overline{\sigma}1,c3), if & x<c3\leq c4 \\ \overline{\mu}_{Gauss}(x,\overline{\sigma}2,c4), if & x>c4,c3\leq c4 \\ 1, if & c3\leq x\leq c4\ and\ c3\leq c4 \\ \overline{\mu}_{Gauss}(x,\overline{\sigma}1,c3)\cdot\underline{\mu}_{Gauss}(x,\overline{\sigma}2,c4), if & c3>c4 \end{cases}$$

$$\overline{\mu}_{Gauss}(x,\overline{\sigma},c) = e^{\frac{-(x-c)^2}{2\overline{\sigma}^2}}$$

$$(8)$$

where $\tilde{X}$ corresponds to type-2 fuzzy sets labelled as $\tilde{L}$, $\tilde{M}$, and $\tilde{H}$. For USC and UFC inputs, formulations based on upper and lower membership functions can be obtained by using Eq. (6). Similarly, membership functions of $\widetilde{NON-SPAM}$ and $\widetilde{SPAM}$ type-2 fuzzy sets for DOSPAM output can be found using Eq. (6) since they are trapezoidal membership functions. In IT2S-FIS, these sets are in singleton form, as in the type-1 fuzzy model, and the most suitable constants for $\widetilde{NON-SPAM}$ and $\widetilde{SPAM}$ fuzzy sets were determined as 0.285 and 0.749, respectively. Figure 8 shows the interval type-2 membership functions defined for the inputs and output of IT2M-FIS.

In IT2M -FIS, the rule form is the same as the rules given in Table 2. The only difference is that instead of type-1 fuzzy sets, Interval type-2 fuzzy sets are used. The inference engine combines the fired fuzzy sets and makes a mapping from the input interval type-2 fuzzy universe to the output interval type-2 fuzzy universe. This process takes place in two stages as in T1M -FIS, implication and aggregation. While the "min" operator was used in the implication phase, the "max" operator was used in the aggregation. However, since the inputs in the antecedent operations of the active rules will cut the interval type-2 fuzzy set as lower membership function and upper membership function at two points, two membership degrees will be produced for each input. Therefore, IT2M -FIS performs implication and aggregation operations based on both memberships. In IT2S -FIS, "prod" is used as the implication operator, and "sum" is used as the aggregation operator in this process. A two-stage process is carried out to reach the final output of DOSPAM, which is represented by a value in the range of 0-1 from the type-2 inference result in IT2M-FIS. In

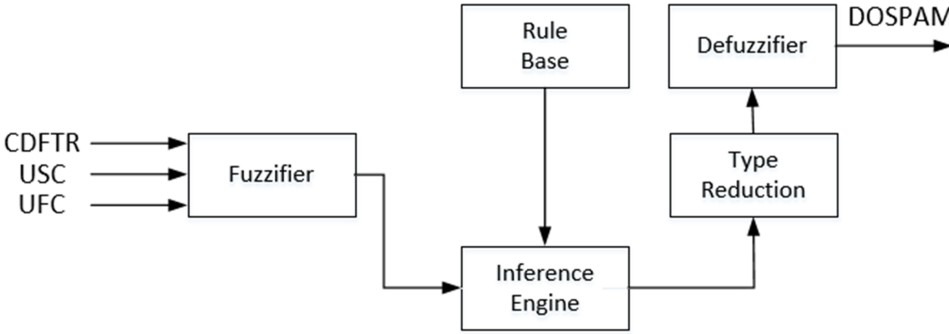

**Figure 7** Block diagram of the IT2M-FIS proposed for calculating the degree of spam.

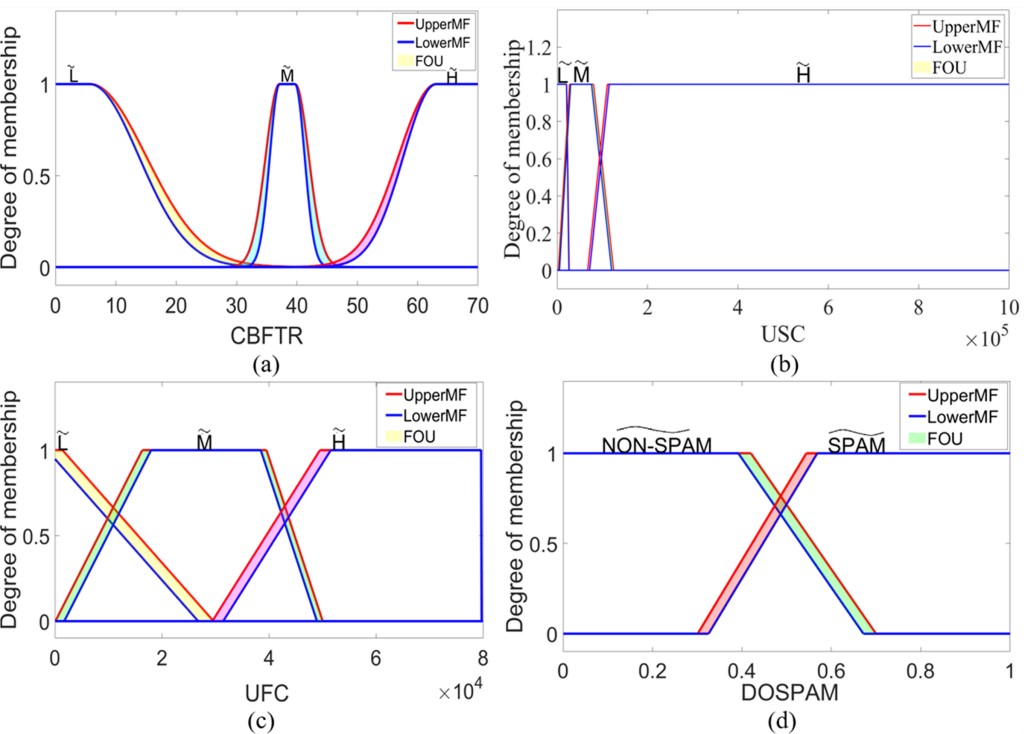

**Figure 8** The membership functions defined for the inputs and outputs of the IT2M-FIS: (A) CBFTR input, (B) USC input, (C) UFC input, and (D) DOSPAM output.

the first step, by applying a series of iterative type reduction methods to the result of the Type-2 fuzzy inference such as Karnik-Mendel (KM), enhanced Karnik-Mendel (EKM), and iterative algorithm with stop condition (IASC), this result is reduced to a type-1 fuzzy set, which is a range with lower limit *cl* and upper limit *cr* (*Atacak & Bay, 2012*; *Ashraf, Akram & Sarwar, 2014a*). In the second step, the final result is obtained by applying the center method (*cl+cr/2*) to this result.

*Classification with ML-based models:* Support vector machine (SVM), Bayes dot machine (BPM), logistic regression (LR), and average perceptron (Avr Prc) methods, which are widely used in the solution of binary classification problems as ML-based models for spam detection in social networks, have been utilised. SVM is a ML algorithm used to solve classification and regression problems (*Noble, 2006*). Based on statistical learning theory, this algorithm can be applied to all linear and nonlinear classification problems. In classification, linear or non-linear (Kernel type) functions are used based on the structure of the process (*Noble, 2006*; *Eliyati et al., 2019*). SVM basically tries to separate two classes with a line or plane. This separation is made based on the elements at the boundary. BPM is an algorithm that uses the Bayesian approach to classify samples given on a network. The basis of algorithm is based on the efficient approximation of the linear classifier to the Bayesian mean over a selected mean classifier or bias point. Since BPM is a Bayesian classification model, it does not tend to overfit the training data (*Herbrich, Graepel & Campbell, 2001*). LR is often the preferred algorithm for linear classification problems rather than regression problems. How the algorithm works is based on fitting a logistic function given as $f\left(\vec{z};\vec{\beta}\right) = \frac{1}{1+e^{-\vec{\beta}^T.\vec{z}}}$ to a labelled dataset. Here $z \rightarrow$ represents the independent variable (input) vector pattern, and $\beta \rightarrow$ represents the regression coefficient matrix (*Hosmer Jr, Lemeshow & Sturdivant, 2013*). The Avr Prc method, which is one of the supervised learning models, creates a simple form of neural network. It requires a dataset with labelled columns for the classification process. The available inputs can be divided into several possible outputs depending on the use of the linear function, and then combined over the weights (*Dineva & Atanasova, 2018*). Our ML-based models in Azure ML platform which is known to enable such algorithms to be implemented are built. The diagram of the models built in the Azure ML platform for the Evaluation phase of the proposed methodology is depicted in Fig. 9.

*Classification with ML-based models:* As it can be seen from the diagram, after the data are transferred to the CSV database as reconciliated data is loaded into the system through the data set module, it is first sent to the filter based feature selection module to carry out the feature selection process. Here, the Fisher score method is used as the filter-based methods in feature selection process. Because it is aimed to keep the number of classifier inputs to a minimum, the "number of desired features" parameters are set to 3 in the module. Then the data with selected features are sent to the Splite data module so that it can be divided into training and test data at specified rates. In this module, the "Fraction of rows in the first output dataset" parameter, which shows the split ratio, is set to 0.7 and 0.8, respectively, depending on the two-stage splitting strategy. In the Splite data module, the data reserved for training is sent from an output to the modules named "Two Class Algorithm Name" for training ML-based models, while the data reserved for testing is sent to the Train model module, which represents the trained model for classification. The result produced by the trained model in the range of 0-1 obtained from the output of this module and the actual class values are converted into a 2-column data over the Score model module. Finally, performance results are obtained by using this data with the Evaluate model module. The adjustment parameters related to the classification algorithms are determined as follows: "number of iterations =5" and "Lambda = 0.002" of the two-class support vector machine

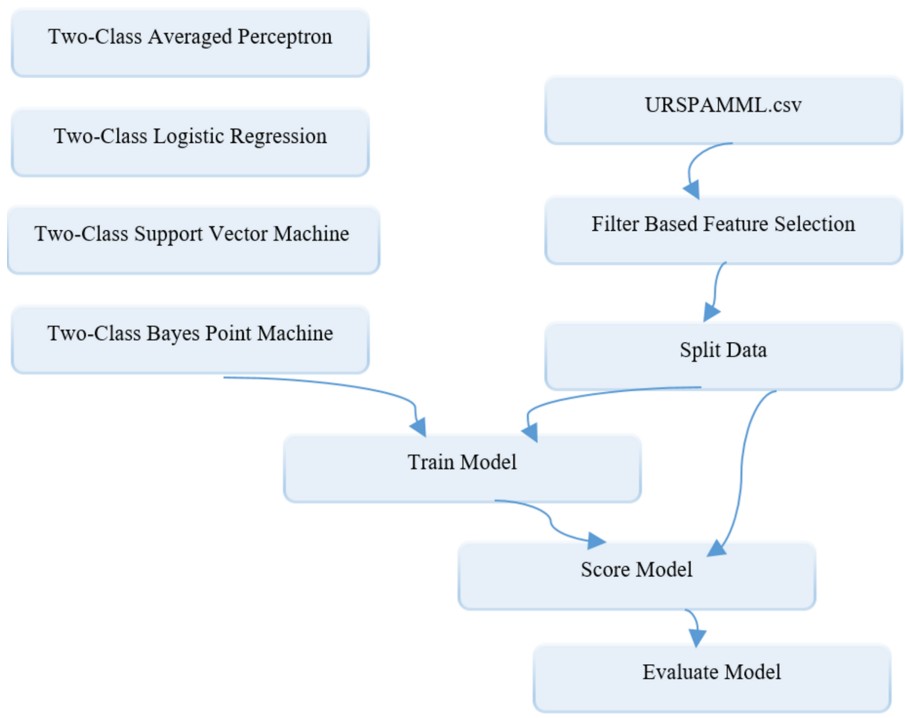

**Figure 9** Schematic representation of ML-based models building in the Azure environment for the assessment phase.

model, "number of training iterations =30" of the two-class Bayes point machine model, "optimization tolerance = 1E-07", "L1 regularization weight = 1", "L2 regularization weight =1" and "memory size for L-BFGS = 20" of two-class logistic regression model, and "learning rate = 0.2" and "maximum number of iterations = 20" of two-class average perceptron model.

## Performance metrics

Until now, a number of performance metrics have been used to accurately evaluate the models, which consist of a combination of different feature selection and classification methods in various applications. These metrics are generally labelled into three categories: threshold, probability, and ranking metrics. Most of the metrics in the mentioned categories use a matrix obtained by cross-validation in measurements. This matrix called as confusion matrix contains four basic concepts related to correctly and incorrectly classified instances to be used in binary classification problems: number of positive instances which are correctly classified (tp), number of negative instances which are correctly classified (tn), number of misclassified positive instances (fp), and number of misclassified negative instances (fn) (*Hossin & Sulaiman, 2015*). In this study, six metrics falling into the threshold and ranking categories were used to measure the common performance of the whole system together with the classification methods during spam detection. The definitions and formulas of these metrics are shown in Table 3.

**Table 3  Metrics used to measure the common performance of the whole system together with the classification methods.**

| Metrics | Formula | Definition |
| --- | --- | --- |
| Accuracy (Acc) | $\frac{tp+tn}{tp+fp+tn+fn}$ | It is a metric that gives the ratio of the number of correctly classified instances among all instances evaluated. |
| Recall (Rec) | $\frac{tp}{tp+fn}$ | It is a metric that measures how many of the actual positives are captured by the learning model in a classification problem given. |
| Precision (Prec) | $\frac{tp}{tp+fp}$ | It is a metric that calculates how many of the positively classified instances are really positive. The ratio of correct predictions to the total correct predictions is called precision. |
| Specificity (Spec) | $\frac{tn}{tn+fp}$ | It is a metric that measures the negative recognition power of the learning model in a binary classification problem. |
| F-score | $2 \times \frac{Rec \times Prec}{Rec+Prec}$ | It also known as F-Measure, which is a metric that gives the harmonic mean of precision (Prec) and recall (Rec) measures. |
| Area Under ROC Curve (AUC) | $AUC = \int TPR.D(FPR)$ | It is a metric that determines the performance measurement of the learning model according to the area under the Receiver Operating Characteristics (ROC) curve, which shows the relationship between the true positive rate (TPR) and the false positive rate (FPR) for different thresholds. |

The first five of the metrics given in the table are in the threshold category, while the AUC is in the ranking category. The metrics in the threshold category decide on the measurement results based on the user-defined threshold values. To calculate the measurement result in this process, it is looked at whether the prediction is below or above the threshold. It doesn't matter how far or close the prediction is to the threshold in terms of the assessment. The obtained measurement result varies from "0" to "1", and its larger values are considered a good performance indicator (*Japkowicz & Shah, 2011*). The ranking metrics are qualified as performance meters based on the ordering of instances according to the output values predicted by the learning models. The performance results for these metrics show how well the positive instances are ordered above the negative ones. The AUC ranking metric that is used in our study measures the classifier performance graphically using the ROC curve which shows the change of true positive rate with false positive rate (*Japkowicz & Shah, 2011*; *Jeni, Cohn & De La Torre, 2013*). As in the metrics in the threshold category, the measurement result of this metric ranges from "0" to "1", and its higher values are interpreted as a good performance indicator.

## RESULTS AND DISCUSSION

Experimental studies were carried out by applying FL and ML-based approaches to the data obtained from Twitter and then, extracted features. IT2M-FIS, IT2S-FIS, T1M-FIS, and T1S-FIS models used as FL-based approaches for spam detection were created in the MATLAB R2021a program and SVM, BPM, LR, and Avr Prc models implemented as ML-based approaches were built in the Microsoft Azure ML platform. For ML-based models, experiments were conducted in two stages, with 70% training-30% test data (split ratio: 0.7) and 80% training-20% test data (split ratio: 0.8). Therefore, experiments for

FL-based models are performed using 30% test data in Stage 1 and 20% test data in Stage 2. FL based approaches are not learning based. However, ML approaches are learning-based. In the experiments, the values given in the "Materials & Methods" section were used as the parameter values of the FL and ML-based models.

## Results for the first stage data split

A training dataset consisting of 4,288 spam data and a test dataset consisting of 1,837 spam data were obtained by applying a 0.70 split ratio to the Twitter dataset including 6,125 spam data. After the ML-based models were trained using the training dataset, experimental results were obtained by applying the test dataset to the trained model. Since FL-based approaches are algorithms that are not based on learning, experimental results were obtained using only the test dataset in spam detection models created with these approaches. The performance results based on confusion matrices and ROC curves of the spam detection models created with FL and ML based approaches under the conditions of 0.70 data split ratio are shown in Fig. 10. Four FL-based spam detection models, including IT2M-FIS, IT2S-FIS, T1M-FIS and T1S-FIS approaches, were applied to the 70% random split dataset, respectively. From the performance results illustrated in Figs. 10A and 10B, it is understood that the IT2M-FIS model among the FL-based models achieved the best performance in terms of accuracy, recall, specificity, precision, F-score, and AUC metrics, with values of 0.955, 0.967, 0.938, 0.957, 0.962, and 0.971, respectively. The T1S-FIS model produced the lowest performance among the FL-based models with a F-score value of 0.928, precision value of 0.951, selectivity value of 0.934, recall value of 0.905, an accuracy value of 0.888 and AUC value of 0.962.

When Fig. 10A is analyzed, it is seen that T1M-FIS and IT2S-FIS fuzzy logic models have very close values to each other. AUC T1M-FIS and IT2S-FIS and accuracyT1M-FIS and IT2S-FIS values are 0.965 and 0.930 respectively in both. Other evaluation metrics have very close values for T1M-FIS and IT2S-FIS FL models. However, IT2M-FIS fuzzy logic model gives higher values than the other three FL models. For IT2M-FIS FL model, it is very high especially when observing at the recall, F-score and accuracy values. However, AUC, Precision and selectivity values are close to each other. On the other hand, T1S-FIS fuzzy logic model gives the lowest values when comparing evaluation metrics. When the recall, F-score and accuracy values are calculated for the T1S-FIS FL model, they give the lowest value among the other three FL models.

It is clearly seen from the graphs in Figs. 10C and 10D that there is no single model that gives the best performance in terms of all metrics in ML-based models. The BMP model provided the best performance in terms of F-score, recall and accuracy from the confusion matrix-based metrics with values of 0.917, 0.912 and 0.903, respectively, while the LR model reached this result with an AUC value of 0.946 in terms of the area under the ROC curve. Among the ML-based approaches, the Avr Prc model has emerged as the model with the lowest performance with F-score value of 0.903, precision value of 0.918, selectivity value of 0.888, recall value of 0.889, accuracy value of 0.888, and AUC value of 0.932.

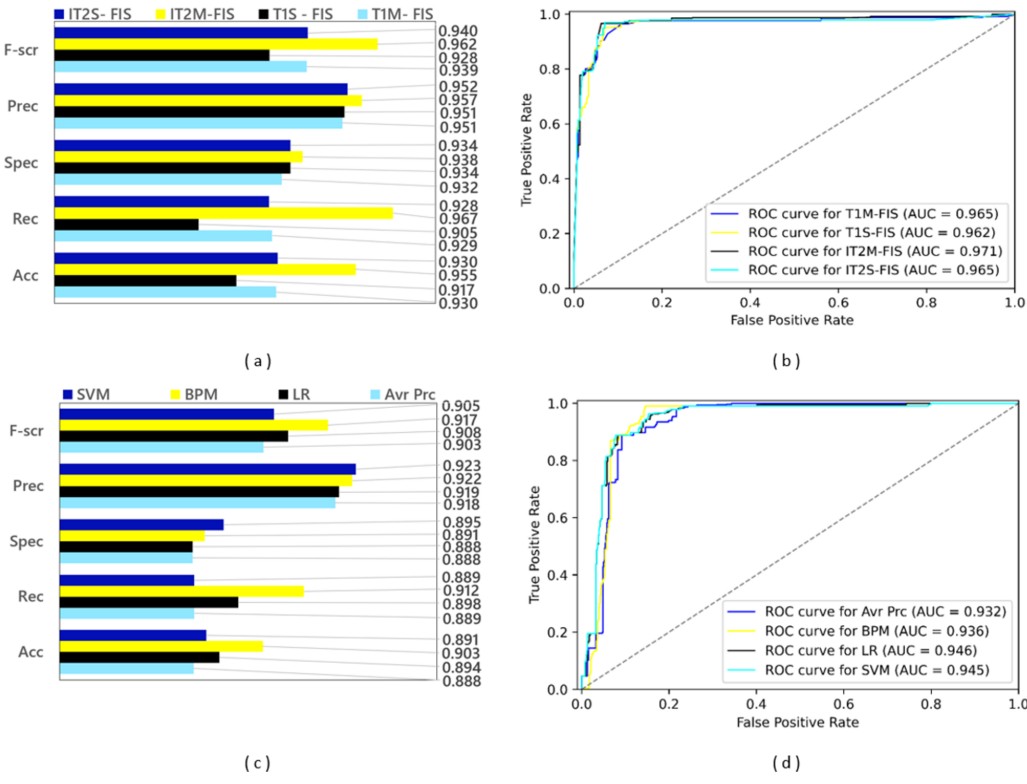

**Figure 10** Performance results of spam detection models created with FL and ML-based approaches at data split conditions of 0.70: (A) confusion matrix metrics for FL-based models, (B) ROC curves for FL-based models, (C) confusion matrix metrics for ML-based models.

According to Fig. 10C, it is very difficult to choose the model that performs best in the ML model. At first glance, the BPM ML model seems to give the best value, but it has the 3rd lowest ACU value (0.936). However, if the AUC value is ignored, it can be said that the BPM ML model is always more successful than the LR ML model. The LR ML model has the highest AUC value (0.946). Avr Prc and LR ML model have lowest selectivity values (0.888). Although it can be said that the Avr Prc ML model has the worst performance when Figs. 10C and 10D are analyzed, it is quite difficult to choose the best performance among the remaining three ML models.

When the spam detection models created with FL and ML-based approaches are compared with each other in general, it is seen that all FL-based models perform much higher than all ML-based models. While the best performance among all spam detection models was achieved with the IT2M-FIS model under the data split conditions of 0.70, the lowest performance was achieved with the Avr Prc model. It is quite difficult to choose the model with the best performance from the ML models, but the most successful model can be easily selected from the FL models.

## Results for the second stage data split

In the second data split step, a split ratio of 0.80 was applied to the Twitter data set, and as a result a training data set containing 4,900 spam data and a test data set containing 1,225 spam data were obtained. The performance results based on confusion matrices and ROC curves obtained by applying FL and ML based spam detection models to these data sets are shown in Fig. 11. Among the FL-based spam detection models, the best performance in terms of all metrics was obtained by IT2M-FIS model with F-score value of 0.959, precision value of 0.946, selectivity value of 0.921, recall value of 0.972, accuracy value of 0.951, and AUC value of 0.973. as can be seen in Figs. 11A and 11B. The lowest performance among the FL-based models was demonstrated by the T1S-FIS model with an accuracy value of 0.919, recall value of 0.925, F-score value of 0.931, and AUC value of 0.962. When the results shown in the graphs in Figs. 11C and 11D are examined, it is seen that there is no single model that yields the best performance in terms of all metrics in the ML-based models, as in the FL-based models. While the BMP model achieved the highest performance with the values of 0.913, 0.942, and 0.927, respectively, in terms of accuracy, recall, and F-score from the confusion matrix-based metrics, the LR model obtained this performance with an AUC value of 0.948 as the largest area under the ROC curve. Among the ML-based approaches, the Avr Prc model had the lowest performance with an accuracy value of 0.902, recall value of 0.916, F-score value of 0.916, and AUC value of 0.932.

While the model with the best values (IT2M-FIS) and the model with the lowest values (T1S-FIS) could be determined among the FL-based models, the most successful model could not be found in ML-based approaches. Regardless of whether the dataset is divided into 70% and 80% test and training datasets, the IT2M-FIS FL method was the most successful model in both cases. In four ML models, the most successful model could not be selected again. The most unsuccessful FL and ML models were the T1S-FIS FL model and Avr Prc ML model in both datasets (70.0% and 80.0% split).

When the performance graphs given in Fig. 11 are examined as a whole, it is understood that FL-based spam detection models outperform ML-based spam detection models in terms of all metrics.

As a result of the transition from the data split conditions of 0.70 to the data split conditions of 0.80, the training data increased and the test data decreased. While this positively affected the performance of ML-based models using the training process, it had a negative impact on the performance of FL-based models with the decrease in test data, except for recall and AUC metrics. However, the reflection of this negative effect on the FL models on the mentioned performance parameters remained at a very low value. Therefore, the IT2M-FIS model provided the highest performance among all the models developed, while the Avr Prc model was the model with the lowest performance in the data split conditions of 0.80, too as in the data split conditions of 0.70. In the literature, there are many studies using different methods and approaches from past to present for spam detection on social media platforms. The current ones of these studies (the ones that have been carried out in recent years) mainly reflect the studies in the field of artificial intelligence. ML-based spam detection models, DL-based spam detection models, community learning-based spam detection models, and FL-based spam detection models

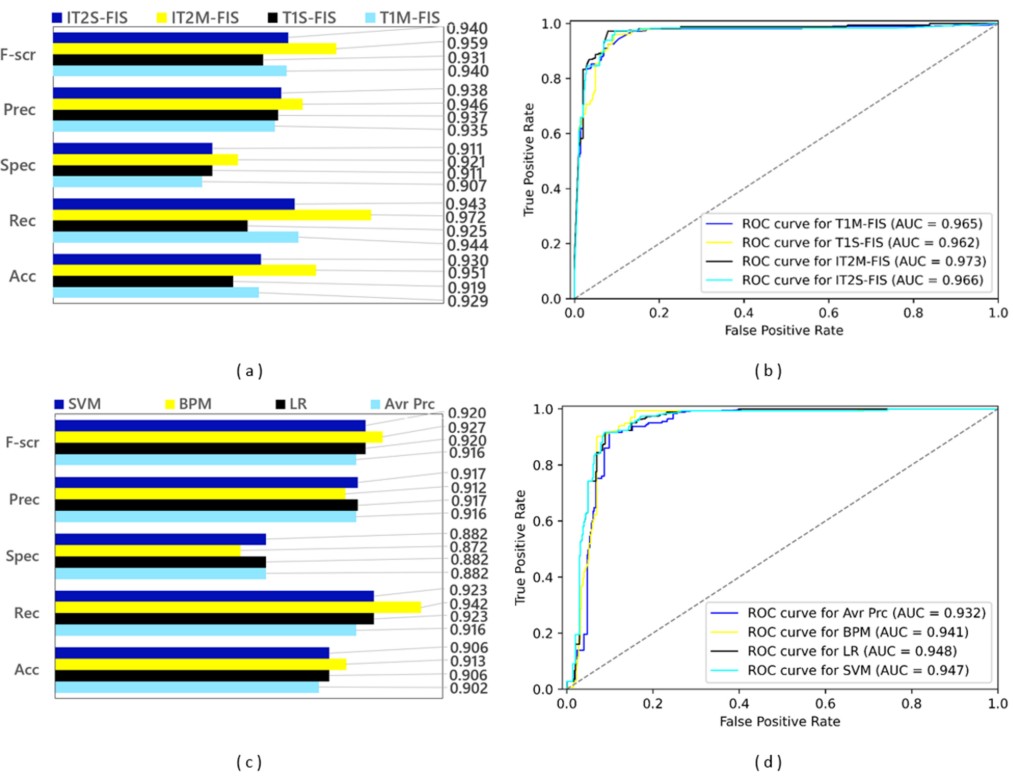

**Figure 11** Performance results of spam detection models created with FL and ML-based approaches at data split conditions of 0.80: (A) confusion matrix metrics for FL-based models, (B) ROC curves for FL-based models, (C) confusion matrix metrics for ML-based models.

represent the models that are relevant to our study. Table 4 provides an overview of the datasets and performance results for these models and the models being proposed.

Like the model we proposed, all of the spam detection models mentioned in Table 4 used the twitter dataset. Methodologies similar to the proposed model and performance metrics are almost the same. In the spam detection models proposed in the literature, fuzzy logic-based ones are not very common. The most important reason for this can be considered as the lack of learning in fuzzy logic-based studies. However, the studies mentioned in Table 4 made evaluations within the framework of fuzzy logic. It has been tried to make comparisons with the most recent fuzzy logic based approaches in the literature.

When Table 4 is examined, it is seen that the CNN and BERT-based TOBEAT approach suggested by *Ouni, Fkih & Omri (2022)* yielded the closest result to our proposed IT2M-FIS-based spam detection model in terms of all performance parameters. However, our model achieved a higher performance of 0.53% in the accuracy metric, 0.82% in the recall metric, 1.65% in the precision metric, and 1.25% in the F-score metric when compared to the TOBEAT approach. The probabilistic clustering approach, which includes rule-based clustering and fuzzy sentiment analysis classifier, suggested by *Ayo et al. (2021)*. to classify hate speech on the Twitter platform, has produced a very close result to our model with

**Table 4  Some studies in the literature on spam detection models recommended.**

| Author | Dataset | Methodology | Performance results |
|---|---|---|---|
| *Ouni, Fkih & Omri (2022)* | SEMCAT-2018 | BERT-and CNN-based TOBEAT | Accuracy 0.9497<br>Recall 0.9588<br>Precision 0.9405<br>F-Measure 0.9495 |
| *Ayo et al. (2021)* | Twitter Dataset (by hatebase.org) | Probabilistic clustering approach: Rule-based clustering+Fuzzy-based sentiment classification | Accuracy: 0.9453<br>Recall: 0.9174<br>Precision: 0.9254<br>F-Measure: 0.9256<br>AUC: 0.9645 |
| *Meriem, Hlaoua & Romdhane (2021)* | Twitter Dataset (SemEval2014 and the Bamman) | Fuzzy Logic-based classification approach | Accuracy 0.909<br>Recall 0.824<br>Precision 0.957<br>F- Measure 0.874 |
| *Liu et al. (2017)* | Twitter Dataset (by Chen) | Ensemble Learning approach: Random oversampling (ROS)+ Random undersampling (RUS)+ Fuzzy-based oversampling (FOS) | For imbalance rate $\gamma = 2$–20<br>Avrg Precision 0.76–0.78<br>Avrg F- Measure 0.76–0.55<br>Avrg False positive rate: 0.11–0.01<br>True positive rate: 0.74–0.43 |
| *Gupta et al. (2018)* | Twitter Dataset (by Twitter API) | Hierarchical Meta-Path Based Approach (HMPS+OSN2): with Feedback + default one-class classifier | Recall: 0.90<br>Precision: 0.95<br>F-Measure: 0.93<br>AUC: 0.92 |
| *Ameen & Kaya (2018)* | Twitter Dataset (by Twitter API) | Deep Learning Approach based on multilayer perceptron (MLP) model | Recall: 0.88<br>Precision: 0.92<br>F-Measure: 0.89 |
| *Madisetty & Desarkar (2018)* | Twitter Dataset 1-HSpam14, 2-1KS10KN | Neural Network-Based Ensemble Approach: 5 CNN models+ feature-based model | Accuracy 0.957<br>$Recall_{1KS10KN}$ 0.867<br>$Precision_{1KS10KN}$ 0.922<br>$F\text{-}Measure_{1KS10KN}$ 0.893<br>$Recall_{HSpam14}$ 0.909<br>$Precision_{HSpam14}$ 0.880<br>$F\text{- }Measure_{HSpam14}$ 0.894<br>AUC 0.964 |
| *Ashour, Salama & El-Kharashi (2018)* | Twitter Dataset (by Lee) | Support Vector Machines (SVM) Random Forests (RF) Logistic Regression (LR) with different character N-grams features | Recall 0.794<br>Precision 0.795<br>F-Measure 0.794 |
| Proposed Model | Twitter Dataset (by Twitter API) | Interval Type-2 Mamdani Fuzzy Inference System (IT2M-FIS) Interval Type-2 Sugeno Fuzzy Inference System (ITS2-FIS) Type-1 Mamdani Fuzzy Inference System (T1M-FIS) Type-1 Sugeno Fuzzy Inference System (T1S) | $Accuracy_{IMT-2FIS}$ 0.955<br>$Recall_{IMT-2FIS}$ 0.967<br>$Precision_{IMT-2FIS}$ 0.957<br>$F\text{- }score_{IMT-2FIS}$ 0.962<br>$AUC_{IMT-2FIS}$ 0.971 |

lower difference values of 0.97% and 0.65%, respectively in terms of accuracy and AUC values. The method used in the study is based on fuzzy logic. Compared to the Recall, precision, and F-score metrics, they are lower than the metrics in the proposed model.

Another approach that only produces results being close to our model in terms of accuracy and AUC metrics, and even slightly higher in accuracy metric (0.2% difference), is the spam detection model proposed by *Madisetty & Desarkar (2018)*. Their neural network-based community model, which combined five CNNs and a feature-based method for spam detection on Twitter, performed 0.7% lower in AUC than our IT2M-FIS-based spam detection model. In fact, the performance of this model in terms of recall, precision and F-score metrics is significantly lower than that of our model. *Meriem, Hlaoua & Romdhane (2021)* used fuzzy logic classification approach in their studies. Although the Precision value (0.957) remains the same as our proposed model value, other performance values are lower. Using the ensemble approach, which is another spam detection method, is included in the modeled fuzzy logic proposed by *Liu et al. (2020)*. The values of precision and F-score shown in Table 4 appear to be lower than the values in our proposed model. The model proposed in this study was also compared with ML methods. IT2M-FIS FL model has been more successful than ML methods. The LR model used in ML methods is also the model used in the study of *Ashour, Salama & El-Kharashi (2018)*. In our study, the LR model was used for comparison. This LR model is more successful than the model in Table 4 (please see Figs. 10C and 11C). Therefore, the IT2M-FIS model we proposed is more successful than the LR model in Table 4. When the other suggested approaches in the table are compared with our model, it is understood that their performance is lower than that of our model. As a result, we can clearly declare that our IT2M-FIS-based model outperforms the approaches in the literature.

## CONCLUSIONS

Nowadays, the popularity of social networking sites such as Twitter has brought about a significant increase in spam accounts on these sites. With this increase, the tools used by social networking platforms to protect their users have become inadequate due to both the diversity of spam and the changes in spam behaviour. In this study, spam detection models based on Type-2 Fuzzy inference and Type-1 fuzzy inference systems were proposed as an effective and powerful approach to overcome the problem mentioned, and the effectiveness of these models was evaluated with performance measures including accuracy, recall, specificity, precision, F-score, and AUC metrics. According to the results obtained from these evaluations, fuzzy logic-based methods for spam detection in social networks give successful results. Although the absence of learning in fuzzy logic-based methods seems to be a disadvantage, FL methods can show high performance compared to supervised ML methods. The results evaluated by the IT2M-FIS FL based model proposed in this study, namely F-score (0.962), precision (0.957), selectivity (0.938), recall (0.967), accuracy (0.955), and AUC (0.971) are important. The data used in the study is a dataset that firstly extracted the features from the raw data obtained through the Twitter API by applying Qui-Squire, chi-square, and the Crowdsourcing Method, and then made it suitable for the inputs of FL-based models through the Data reconciliation process. Four different FL-based spam detection models, namely IT2M-FIS, IT2S-FIS, T1M-FIS, and T1S-FIS, were applied to this dataset for classification purposes. In order to evaluate

the performance of the proposed spam detection models, the same dataset was also tested with four basic ML-based spam detection models built in the Azure ML platform, including SVM, BPM, LR, and Avr Prc methods. Experiments under 70% and 80% data split conditions demonstrated that FL-based models performed a better performance than ML-based models. Comparing the proposed method in this study with different ML learning methods with the same dataset provides support that increases the importance of the study. When the FL-based models were evaluated among themselves, it was seen that the models with Mamdani inference system outperformed the models with Sugeno inference system. The IT2M-FIS-based spam detection model exhibited the highest performance among the implemented models in terms of all metrics. This model also had a higher performance than the studies in the literature. Studies have shown us that it is necessary to pay attention to the security of smart mobile devices that have internet access and enable the use of social networks. The social account has not problem, but the maliciousness of the device used can significantly affect social network security. In our future studies, it is important to focus on mobile internet devices and cyber security awareness, and combining different spam classification methods and applying hybrid new methods will yield good results.

### Funding
The authors received no funding for this work.

### Competing Interests
The authors declare there are no competing interests.

### Author Contributions

- İsmail Atacak conceived and designed the experiments, performed the experiments, analyzed the data, performed the computation work, prepared figures and/or tables, authored or reviewed drafts of the article, determined the algorithm used in the study, checked the models used and observed the result obtained and wrote the manuscript. He also performed the experiments, analyzed the data, performed the computation work, prepared figures and/or tables, authored or reviewed drafts of the article, and approved the final draft, and approved the final draft.
- Oğuzhan Çıtlak conceived and designed the experiments, performed the experiments, analyzed the data, performed the computation work, prepared figures and/or tables, authored or reviewed drafts of the article, obtained the dataset on the social network, performed the models, analyzed the results and wrote the manuscript as well. He performed the computation work, reviewed drafts of the article, prepared figures and/or tables, and approved the final draft, and approved the final draft.
- İbrahim Alper Doğru conceived and designed the experiments, performed the experiments, analyzed the data, performed the computation work, prepared figures and/or tables, authored or reviewed drafts of the article, checked the studies in

the literature the manuscript, determined the algorithms used and played a role in the creation of the model. In addition, he designed the experiments, performed the experiments, analyzed the data, authored or reviewed drafts of the article, and approved the final draft, and approved the final draft.

## Data Availability

Raw data and code are available in the Supplemental Files.

1. Python codes used to obtain the dataset via Twitter.

2. Taxonomies of the dataset used.

3. API IDs of the account used to obtain the dataset in Twitter

4. Mamdani 1-2 and Sugeno 1-2 fis files of the proposed Fuzzy Logic model/codes files run on MATLAB.

5. Reconciliatedspam dataset file

6. Criterion ranges file used in machine learning

7. Spam datasets

8. Real Twitter dataset

9. Fuzzymatlab code

## Supplemental Information

Supplemental information for this article can be found online at http://dx.doi.org/10.7717/peerj-cs.1316#supplemental-information.

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
