# Peer review of "Application of interval type-2 fuzzy logic and type-1 fuzzy logic-based approaches to social networks for spam detection with combined feature capabilities"

_PeerJ Computer Science, doi:10.7717/peerj-cs.1316_

## Round 0.1 · original submission · Major Revisions

Please revise your paper according to the reviewer's comments.
Thanks

Reviewer 1 ·

Basic reporting

1-The abstract is not convincing, it should be refined to precisely illustrate what authors have done in this paper. The abstract must be a concise yet comprehensive reflection of what is in your paper. Remember that reader want to know: 1-what is the problem. 2- why the problem is relevant 3- wants an overview of your approach. 4-need to know the results.
1-Manuscript needs a good introduction, the introduction section of the manuscript is weak, authors are advised to improvise the introduction section.
2-In the Introduction part, the new features of the proposed method and the main advantages of the results over others should be clearly described.
3-An introduction should clearly highlight the motivation, problem statement, the objective of the paper, gap in the existing research and the novelty of the conducted research.

Experimental design

The article need to be revised with more experimentation by comparison relevant approaches and algorithms. More recent article should be considered for providing the proposed approach effectiveness.

Validity of the findings

1. Result and Discussion section is inadequate. Need more attention and better explanation.
2. When I checked the results, I noticed that there were mistakes, please recheck

Additional comments

1. English language needs to be improved significantly.
2. I suggest extending the conclusions section to focus on the results you get, the method you propose, and their significance.
3. Cite related work:
A fuzzy climate decision support systems for tomatoes in high tunnels, International Journal of Fuzzy Systems, 19(3)(2017):751-775
Fuzzy decision support system for fertilizer, Neural Computing & Applications, 25(6)(2014) 1495-1505
Type-II fuzzy decision support system for fertilizer, Scientific World Journal, 2014 (2014), Article ID 695815, 9 pages

Reviewer 2 ·

Basic reporting

The manuscript titled "Application of interval type-2 fuzzy logic and type-1 fuzzy logic-based approaches to social networks for spam detection with combined feature capabilities" is well written, well presented and is scientifically correct.

Experimental design

Satisfactory.

Validity of the findings

Valid.

Additional comments

I recommend that this manuscript can be reconsidered after the following major revisions:
1. The abstract of the manuscript is not convincing, regarding expressing the proposed problem. It should be revised.
2. Language quality needs improvement.
3. Figure 1 is almost unreadable, figure 9 is fade and on page 30 figure 11 have some errors. These issues need to be addressed.
4. Cite proper references for basic definitions and results.
6. Add future studies in conclusion section.
7. Write all references on the same pattern.

Reviewer 3 ·

Basic reporting

See below

Experimental design

See below

Validity of the findings

See below

Additional comments

The authors proposed a new application of interval type-2 fuzzy logic and type-1 fuzzy logic-based approaches to social networks for spam detection with combined feature capabilities.
The paper is in good form, well organized, well written and mathematically correct up to the best of my knowledge. I recommend that this article can be published. However, I have some suggestions given below:
1. Rewrite the abstract and the introduction sections of the paper, it should provide problem statement with contributions.
2. The symbols and notations used in the paper should be simple.
3. Each new definition should be followed by an example.
4. There is need for more elaboration from some of the references cited in the introduction.
5. English language and style are fine, some minor spell check is required.
4. Include some related references of recently published in this journal.
5. The paper should be in the format of this journal.
6. The authors should add future work in the light of these related topics. The future research scope is not clearly mentioned in the conclusion section. More specific future research directions are welcome.
7. Some comments on literature are not up to date. Please read and update related literature;
Pythagorean fuzzy prioritized aggregation operators with priority degrees for multi-criteria decision-making; A study on weighted aggregation operators for q-rung orthopair m-polar fuzzy set with utility to multistage decision analysis; Group decision-making using complex $q$-rung orthopair fuzzy Bonferroni mean.
8. The authors should improve introduction with recent advances in fuzzy modeling, linear Diophantine fuzzy sets, etc.
9. A comparative analysis of the suggested technique with existing approaches is missing. Add advantages and limitations of proposed work.

---

## Round 0.2 · Major Revisions

The authors only cited the references and did not focus on the quality of the work and the reviewer's comments. Please revise your paper carefully and highlight all the changes (blue) in your paper.

Reviewer 1 ·

Basic reporting

This is a nice work.

Experimental design

Nice.

Validity of the findings

Nice.

Additional comments

.

Reviewer 3 ·

Basic reporting

This paper is revised according to the reviewer's suggestions.
This paper may be accepted.

Experimental design

This paper is revised according to the reviewer's suggestions.
This paper may be accepted.

Validity of the findings

This paper is revised according to the reviewer's suggestions.
This paper may be accepted.

Additional comments

This paper is revised according to the reviewer's suggestions.
This paper may be accepted.

---

## Round 0.3 · accepted · Accept

The paper is well-revised and now is ok for publication. Thanks